# New Analytical Approach for the Alignment of Different HE4 Automated Immunometric Systems: An Italian Multicentric Study

**DOI:** 10.3390/jcm11071994

**Published:** 2022-04-02

**Authors:** Antonio Angeloni, Corrado De Vito, Antonella Farina, Daniela Terracciano, Michele Cennamo, Rita Passerini, Fabio Bottari, Annalisa Schirinzi, Roberto Vettori, Agostino Steffan, Valerio Mais, Ferdinando Coghe, Luigi Della Corte, Giuseppe Bifulco, Valentina Baccolini, Elena Berardelli, Giuseppe Migliara, Emanuela Anastasi

**Affiliations:** 1Department of Experimental Medicine, Sapienza University of Rome, Viale Regina Elena 324, 00161 Rome, Italy; antonio.angeloni@uniroma1.it (A.A.); antonella.farina@uniroma1.it (A.F.); elena.berardelli@uniroma1.it (E.B.); 2Department of Public Health and Infectious Diseases, Sapienza University of Rome, 00185 Rome, Italy; corrado.devito@uniroma1.it (C.D.V.); valentina.baccolini@uniroma1.it (V.B.); giuseppe.migliara@uniroma1.it (G.M.); 3Department of Translation Medical Sciences, University of Naples Federico II, 80131 Naples, Italy; daniela.terracciano@unina.it (D.T.); michele.cennamo@unina.it (M.C.); 4Division of Laboratory Medicine, European Institute of Oncology, IRCCS, 20141 Milan, Italy; rita.passerini@ieo.it (R.P.); fabio.bottari@ieo.it (F.B.); 5Clinic Pathology Unit, University Hospital, Policlinico di Bari, 70124 Bari, Italy; schianna@gmail.com; 6Immunopathology and Cancer Biomarkers, Centro di Riferimento Oncologico di Aviano (CRO Aviano), IRCCS, National Cancer Institute, 33081 Aviano, Italy; rvettori@cro.it (R.V.); asteffan@cro.it (A.S.); 7Department of Surgical Sciences, University of Cagliari Medical School, 09042 Cagliari, Italy; vmais@unica.it; 8Department of Laboratory Medicine, University Hospital, 09123 Cagliari, Italy; coghe.f@tiscali.it; 9Department of Neuroscience, Reproductive Sciences and Dentistry, School of Medicine, University of Naples Federico II, 80131 Naples, Italy; luigi.dellacorte@unina.it (L.D.C.); giuseppe.bifulco@unina.it (G.B.)

**Keywords:** biomarkers, human epididymal secretory protein 4 (HE4), multicentric study, ovarian cancer

## Abstract

Human epididymal secretory protein 4 (HE4) elevation has been studied as a crucial biomarker for malignant gynecological cancer, such us ovarian cancer (OC). However, there are conflicting reports regarding the optimal HE4 cut-off. Thus, the goal of this study was to develop an analytical approach to harmonize HE4 values obtained with different laboratory resources. To this regard, six highly qualified Italian laboratories, using different analytical platforms (Abbott Alinity I, Fujirebio Lumipulse G1200 and G600, Roche Cobas 601 and Abbott Architett), have joined this project. In the first step of our study, a common reference calibration curve (designed through progressive HE4 dilutions) was tested by all members attending the workshop. This first evaluation underlined the presence of analytical bias in different devices. Next, following bias correction, we started to analyze biomarkers values collected in a common database (1509 patients). A two-sided *p*-value < 0.05 was considered statistically significant. In post-menopausal women stratified between those with malignant gynecological diseases vs. non-malignant gynecological diseases and healthy women, dichotomous HE4 showed a significantly better accuracy than dichotomous Ca125 (AUC 0.81 vs. 0.74, *p* = 0.001 for age ≤ 60; AUC 0.78 vs. 0.72, *p* = 0.024 for age > 60). Still, in post-menopausal status, similar results were confirmed in patients with malignant gynecological diseases vs. patients with benign gynecological diseases, both under and over 60 years (AUC 0.79 vs. 0.73, *p* = 0.006; AUC 0.76 vs. 0.71, *p* = 0.036, respectively). Interestingly, in pre-menopausal status women over 40 years, HE4 showed a higher accuracy than Ca125 (AUC 0.73 vs. 0.66, *p* = 0.027), thus opening new perspective for the clinical management of fertile patients with malignant neoplasms, such as ovarian cancer. In summary, this model hinted at a new approach for identifying the optimal cut-off to align data detected with different HE4 diagnostic tools.

## 1. Introduction

Ovarian, endometrial and uterine cancer are the most frequent gynecologic malignancies affecting women worldwide [1,2]. These tumors, in the early stages of the disease, are often asymptomatic or show modest clinical manifestations. These peculiar aspects of malignancies often give rise to a delay in diagnosis, significantly worsening the prognosis and reducing life expectancy. This is especially true for ovarian cancer (OC), the 5th worldwide leading cause of death for cancer in women, where a late diagnosis, in more than 70% of cases, results in a low five-year survival rate [3]. To this regard, it has been demonstrated that accurate and early identification of a malignant pelvic mass is essential for the appropriate triage, referral, and subsequent care of women diagnosed with an ovarian malignancy and to improve patient outcomes [4,5,6].

Due to the limited cost and invasiveness, circulating biomarkers acquired relevance in the relapse of several malignancies, thus leading to the appropriate management and therapy of OC patients [7].

Currently, oncology research has focused attention onto the identification of highly specific biological markers as indicators of disease activity, but also as prognostic and predictive factors of survival, recurrence, and treatment response in female patients [8]. According to the international guidelines, mucin Ca125 is considered the gold standard tumor marker in OC; however, its clinical use in the management of this malignancy is limited since this biomarker is also increased in other cancers, as well as in benign gynecologic diseases [7,9]. Hence, there is a need to strengthen the knowledge on tumor markers detectable in serum samples available to date. To this regard, human epididymal secretory protein 4 (HE4) has been recently accepted by the US Food and Drug Administration (FDA) as a monitoring tool for the management of patients with OC, thus becoming a useful marker of clinic outcome [10].

In the last decade, multiple algorithms have been developed to help clinicians with the differential diagnosis between benign or malignant pelvic masses. Among them, the Risk of Ovarian Malignancy Algorithm (ROMA) is a logistic regression algorithm that uses HE4 and Ca125 levels, along with the patient’s menopausal status, to categorize patients into high- and low-risk probabilities that a malignancy will be found in a patient with a pelvic mass. Despite the high sensitivity (94%) and the high negative predictive value (NPV) of 99%, this algorithm showed a specificity of 75% for predicting the presence of epithelial ovarian cancer in women presenting with a pelvic mass [11,12,13]. In this contest, HE4 alone has been considered more reliable than the ROMA algorithm [7,14]. These observations point out an important issue: the use of HE4, as a circulating biomarker for the diagnosis of OC, alone or in combination with other biomarkers, must be reliable and standardized.

The first suitable HE4 assay was the manual enzyme immunometric assay (EIA), developed by Fujirebio [15]. Early studies performed with manual assay suggested the prominent role of HE4 in the clinical setting of OC, even though manual tasks were considered unfavorable due to time-consuming, poor precision, and low standardization. Because of this, numerous modular immunometric systems have been developed. To date, automation is considered one of the most important breakthroughs in diagnostics since it improves efficiency, accuracy, reproducibility, standardization, quality and safety of laboratory testing. Moreover, human errors, analysis reporting time and costs are drastically reduced [16]. Currently, different HE4 analyzers based on different immunosensor (i.e., chemiluminescence enzyme immunoassay CLEIA, electrochemiluminescence ECLIA, chemilunescent microparticle capture immunoassay CMIA) are available. However, these systems report dishomogeneous results arising from the lack of data harmonization [17].

In this scenario, the goal of this study was to develop an analytical approach harmonizing HE4 values obtained with different laboratory resources. To this aim, six highly qualified Italian laboratories have joined this project. In the first part of our study, we started to align the different system referring to a standard curve, common to all diagnostic instrumentation. Next, following bias correction, we analyzed sera from 1509 patients.

## 2. Materials and Methods

### 2.1. Study Design

A retrospective observational multicentric study involving six high qualified automated laboratories was planned by The Tumor Markers Laboratory of Azienda Ospedaliera Policlinico Umberto 1 Sapienza University.

Six laboratories joined the project using different analytical platforms:

Lab1: *Azienda Ospedaliera*-*Universitaria Policlinico Umberto I, Rome* → LUMIPULSE G1200II, competitive chemiluminescent enzyme immunoassay (CLEIA);

Lab2: *CRO Aviano (Oncological Reference Center of Aviano, PN**)* → COBAS E601 Electrochemiluminescence (ECLIA);

Lab3: *Policlinico Di Bari (BARI)* → ARCHITECT chemilunescent immunoassay with capture of microparticles (CMIA);

Lab4: *IEO (European Institute of Oncology, MI)* → ALINITY I, chemilunescent microparticle capture immunoassay (CMIA);

Lab5: *Azienda Ospedaliera*-*Universitaria*
*dI Cagliari (CA)* → LUMIPULSE G600II, competitive chemiluminescent enzyme immunoassay (CLEIA);

Lab6: *Azienda Ospedaliera*-*Universitaria Federico II (NA)* → LUMIPULSE G600II, competitive chemiluminescent enzyme immunoassay (CLEIA).

### 2.2. Reference Calibration Curve

As reported above, each laboratory used different immunoanalysers and immunometric techniques for the determination of Ca125 and HE4:

Lab1, project representative, built a ten-point concentration standard curve through progressive HE4 dilutions, from a minimum value of 30.4 to a maximum of 1075.4 pmol/L. The standard curve was shared with each center. All the laboratories analyzed the samples blindly and in duplicate, using the immunometric technique and instrumentation in their possession. Data concerning the analytical session, calibration and quality controls were reported in a document and then sent to Lab1.

### 2.3. Sample Size Calculation

The calculation of the sample size was performed assuming a prevalence of malignancy among women with ovarian tumors of 20% [18] and an expected sensitivity of the test of at least 85%, accepting a 95% confidence interval with a width not exceeding 0.05. The resulting sample size was of 980 patients.

### 2.4. Database Population

A common database of 1509 women was created providing clinical information and biochemical parameters, including HE4 and Ca125 values.

Inclusion criteria: aged between 18 and 70 years; for all patients in this study, a pelvic mass was confirmed by imaging (i.e., ultrasound, computed tomography, or magnetic resonance), documenting a pelvic mass or ovarian cyst thirty days prior to surgery. All patients underwent surgery for a subsequent tumor histological evaluation and full surgical staging for women diagnosed with a malignancy.

Exclusion criteria: pregnant patients and women under the age of 18 were excluded from the trial. Patients with a prior history of a bilateral salpingo-oophorectomy were ineligible for the trial. Patients with creatinine levels ≤ 0.3 mg/dL and >3 mg/dL, with ambiguous gynecological diagnosis, without reported age and classified as in premenopausal stage with age > 55 were excluded.

Enrolled population was subdivided as follows:(a)Healthy women: Healthy Blood donors without gynecological diseases have been recruited from the Transfusional Unit of the Policlinico Umberto I of Rome. Healthy women were recruited according to the criteria and physical requirements for the selection of blood and blood-components donors following “the disposition related to the quality requirements and safety of blood and its components”, published in Gazzetta Ufficiale no.69 (2 November 2015).(b)Benign gynecological diseases: Ovarian cyst, endometriosis, uterine and vaginal fibroma, hemorrhagic corpus luteum, endometrial hyperplasia, endometrial polyp, salpingitis, chronic cervicitis, Brenner’s tumor, cystadenoma of the ovary, vesicular mola. All patients included in this group had imaging and histological confirmation of these clinical status.(c)Malignant gynecological diseases: Ovarian cancer. All patients, following surgery, have had confirmation of the malignancy of the neoplasm and relative staging.

### 2.5. Serum Sample

For healthy women, the sample was taken at the time of the pre-donation interview once the subject’s state of health was confirmed.

For patients with malignant and non-malignant gynecological pathologies, samples were taken at the time of diagnosis, before any pharmacological and surgical treatment. Each patient had levels of HE4, Ca125, and other biomarkers.

All laboratories applied the same standard protocol to collect sera samples. Blood samples were collected in a red-top vacutainer, clotted 60–90 min, and centrifuged for 10 min at 1300× *g*. The serum fractions we aliquoted and stored at −80 °C until analysis.

### 2.6. HE4 Immunoassay

Immunoanalyzers and Immunometric Techniques.

*LUMIPULSE G1200II and G600II (CLEIA) chemiluminescence*. The HE4 Lumipulse G Fujirebio assay uses a chemiluminescent competitive enzyme immunoassay (CLEIA), conducted on the LUMIPULSE G 1200 II analyzer and LUMIPULSE G600II. It is a two-step sandwich test that uses two monoclonal antibodies, 2H5 MAb, which coats paramagnetic microparticles and 12A2 MAb, labeled with alkaline phosphatase. All necessary components are packaged in a single ready-to-use cartridge.The manufacturer declares a total coefficient of variation (CV) ≤ 3.2%.Internal Controls → Three different internal controls (*Level1* 48.3–89.8; *Level2* 398–738; specimen pool) were processed to test the correct functionality of the analyzer.Calibration → A two-point master curve with a range between 0 and 1500 pmol/L was used for calibration.LoD 3.5 pmol/L, LoQ 3.5 pmol/L and linearity was between 20.0 and 1500 pmol/L.*COBAS E601 (Roche Diagnostics, Manheim, Germany) Electrochemiluminescence (ECLIA).* The Elecsys HE4 test is an electrochemiluminescence immunoassay performed on Cobas E601. The development of ECL immunoassays is based on the use of a ruthenium and tripropylamine (TPA) complex. IT is based on a one-step sandwich principle, using a biotinylated 2H5 MAb and the 12A2 MAb labeled with a ruthenium complex for sandwich detection. The latter MAb binds the amino terminal domain of the whey acidic four-disulphide core (WFDC) of the HE4 protein. The chemiluminescence reaction for detecting the reaction complex is initiated by applying a voltage to the sample solution, which results in a precisely controlled reaction.The manufacturer declares a CV ≤ 10%.Internal Controls → Three different internal controls (high, medium and low concentration levels) were processed to test the correct functionality of the analyzer.Calibration → A two-point master curve with a range between 0 and 1500 pmol/L was used for calibration.LoD 5.00 pmol/L, LoQ 20.0 pmol/L and linearity was between 15.0 and 1500 pmol/L.*ARCHITECT (Abbott Diagnostics, North Chicago, IL, USA) (CMIA) chemiluminescence*. The assay HE4 is a microparticle capture chemiluminescent immunoassay (CMIA), performed on the Architect i2000 platform. It is a two-step non-competitive assay performed using two monoclonal antibodies (MAb) directed against two different epitopes of the carboxy terminal domain of the whey acidic four-disulphide core (C-WFDC) of the HE4 protein, 2H5 MAb, capture antibody and 3D8 MAb detection antibody. The chemiluminescent reaction is detected as a relative light unit (RLU). The amount of He4 in the sample and the detected RLUs are directly related parameters.The manufacturer declares a CV ≤ 10%.Internal Controls → Three different internal controls (high, medium and low concentration levels) are processed to test the correct functionality of the analyzer.Calibration → A six-point calibration with a range between 0 and 1500 pmol/L was used for calibration. The calibrators for the assay (in liquid form ready to use lot n.88K10514) were made volumetrically.LoD < 15.0 pmol/L, LoQ < 20.0 pmol/L and linearity was between 20.0 and 1500 pmol/L.*ALINITY I (LAB4) (Abbott Laboratories, IL, USA) (CMIA) chemiluminescence*: The HE4 test is a microparticle capture chemiluminescent immunoassay (CMIA), performed on the Anility I analyzer.The sample and the paramagnetic microparticles coated with antibodies anti-HE4 2H5 are dispensed together and incubated. The HE4 antigen in the sample binds to microparticles coated with anti-HE4 antibodies. The mixture is washed. Acridinium-labeled anti-HE4 3D8 antibody conjugate is added to form a mixture reaction, which is then incubated. After a wash cycle, pre-trigger and trigger solutions are added. The resulting chemiluminescent reaction is measured in relative light units (RLUs). There is a direct link between the amount of HE4 antigen in the sample and the RLUs measured by the optical system.The manufacturer declares a total CV ≤ 4.9%. Two different internal controls (low and medium) were processed in order to test the correct functionality of the test and analyzer: low control ranged from 35–65 pmol/L and medium control was between 122.5–227.5 pmol/L.Calibration of HE4 assay was performed with six calibrators, with a concentration ranging between 0 and 1500 pmol/L.LoD 0.5 pmol/L and LoQ 2.0 pmol/L. Linearity of the assay was between 20.0 and 1500.0 pmol/L.

### 2.7. Ca125 Immunoassay

*LUMIPULSE G1200II and G600II (CLEIA) chemiluminescence*. The Ca125 Lumipulse G Fujirebio assay uses a CLEIA Chemiluminescent Competitive Enzyme Immunoassay, conducted on the LUMIPULSE G 1200 II analyzer and LUMIPULSE G600II. It is a two-step sandwich test that uses two monoclonal antibodies, OC125 MAb, which coats paramagnetic microparticles and M11 MAb, labeled with alkaline phosphatase. All necessary components are packaged in a single ready-to-use cartridge. The manufacturer declares a total CV ≤ 4%.Internal Controls → Three different internal controls were processed to test the correct functionality of the analyzer.Calibration → A two-point master curve with a range between 0 and 1000 U/mL was used for calibration.LoD 0.657 U/mL, LoQ 0.657 U/mL and linearity was between 13.9 and 1000 U/mL.*COBAS E601 (Roche Diagnostics, Manheim, Germany) Electrochemiluminescence (ECLIA*). The Elecsys Ca125 test is an electrochemiluminescence immunoassay performed on Cobas E601. The development of ECL immunoassays is based on the use of a ruthenium and tripropylamine (TPA) complex. IT is based on a one-step sandwich principle, using a biotinylated M11 MAb and the OC125 MAb labeled with a ruthenium complex for sandwich detection. The chemiluminescence reaction for detecting the reaction complex is initiated by applying a voltage to the sample solution, which results in a precisely controlled reaction. The manufacturer declares a CV ≤ 6.5%.Internal Controls → Three different internal controls (high, medium and low concentration levels) were processed to test the correct functionality of the analyzer.Calibration → A two-point master curve with a range between 0 and 5000 U/mL was used for calibration.LoD 1.2 U/mL, LoQ 2.0 U/mL and linearity was between 0.6–5000 U/mL.*ARCHITECT (Abbott Diagnostics, North Chicago, IL, USA) (CMIA) chemiluminescence*. The Assay Ca125 is a microparticle capture chemiluminescent immunoassay (CMIA), performed on the Architect i2000 platform. In the first step, the sample and OC 125 MAb coated paramagnetic microparticles are combined. OC 125 defined antigen present in the sample binds to the OC 125 coated microparticles. After washing, M11 MAb acridinium-labeled conjugate is added in the second step. The chemiluminescent reaction is detected as a relative light unit (RLU). The amount of Ca125 in the sample and the detected RLUs are directly related parameters. The manufacturer declares a CV ≤ 10%.Internal Controls → Three different internal controls (high, medium and low concentration levels) were processed to test the correct functionality of the analyzer.Calibration → A six-point calibration with a range between 0 and 1000 U/mL was used for calibration.LoD ≤ 1.0 U/mL, LoQ ≤ 1.0 U/mL and linearity was between 1.0 and 1000 U/mL.*ALINITY I (LAB4) (Abbott Laboratories, IL, USA) (CMIA) chemiluminescence*. The Ca125 test is a microparticle capture chemiluminescent immunoassay (CMIA), performed on the Anility I analyzer. The assay uses paramagnetic microparticles coated with OC 125 monoclonal antibody to bind the molecules containing the antigen OC 125. These antigens are quantified using acridinium-labeled M11 antibody. The resulting chemiluminescent reaction is measured in relative light units (RLUs). There is a direct link between the amount of Ca125 antigen in the sample and the RLUs measured by the optical system. The manufacturer declares a total CV ≤ 4.8%.Internal controls (low and medium) were processed in order to test the correct functionality of the test and analyzer.Calibration of Ca125 assay was performed with six calibrators with a concentration range between 0 and 1000 U/mL.LoD 0.3 U/mL and LoQ 0.6 U/mL. Linearity of the assay was between 1.1 and 1000.0 U/mL.

### 2.8. Statistical Analisys

Descriptive statistics were reported using mean and standard deviation for continuous variables, and using absolute and relative frequencies for dichotomous and categorical variables. To assess for fixed and proportional systematic biases in HE4 measurement between the various laboratories and Lab1 (referring laboratory), the calibration curve results measurements were used to build ordinary least product linear regression models [19]. Briefly, the alpha coefficients whose CI does not contain 0 indicates the presence of a significant fixed bias, while the beta coefficients whose CI does not contain 1 indicates the presence of a significant proportional bias. Measurement from laboratories with significant biases were than adjusted using the appropriate regression equation to correct for such biases.

Factors influencing the accuracy of HE4 in classifying oncology disease were investigated through a parametric analysis of a receiver operating characteristic (ROC) curve, using a two-step ROC regression model with bootstrap sampling [20]. Variables initially included in the model were age (continuous), menopausal state (dichotomous) and creatinine levels (continuous) for both the control model and for the ROC estimation model, while the hospitals of origin (categorical) were used as a cluster variable for the bootstrap sampling. Due to high correlation between age and menopausal state, only the latter was kept in the model.

Cut-off level for dichotomization of HE4 was obtained by maximizing the Youden index (sum of sensitivity and specificity) against histological classification (malignant gynecological diseases vs. non-malignant gynecological diseases and healthy women, or vs. non-malignant gynecological diseases alone), rounding the result. For Ca125 dichotomization, a standard cut-off of >35 UI/mL was used. ROC curves of dichotomous HE4 and dichotomous Ca125 against histological classification were calculated, and the respective area under the curve (AUC) confronted using the test for equality. A sensitivity analysis was performed, considering as non-malignant gynecological diseases only ovarian, annexal and peritoneal lesions, and endometriosis.

All analyses were performed using STATA 17.0 (StataCorp LLC., 4905 Lakeway Drive, College Station, TX, USA). A two-sided *p*-value < 0.05 was considered statistically significant.

## 3. Results

In total, 1509 subjects were enrolled. The baseline characteristics of the population are summarized in Table 1.

### 3.1. Systematic Biases in HE4 Measurement

Results of the measurement of the calibration curve are reported in Table 2. Lab2 and Lab3 both showed significant fixed and proportional bias, Lab4 had a significant fixed bias, Lab5 had a significant proportional bias, while Lab6 was the only center free from systematic bias.

### 3.2. Analysis of Factors Influencing HE4 Accuracy

In the ROC regression analysis, being in post-menopausal status significantly increased the AUC of HE4 incorrectly classifying the patients (β 0.52, 95% CI 0.31–0.73), while increasing creatinine levels reduced the accuracy (β −1.02, 95% CI −1.74–−0.29) (Table 3). No factors showed a significant effect on HE4 levels in the control model.

### 3.3. Accuracy of Dichotomous HE4

Given the results of the ROC regression analysis, pre-menopausal and post-menopausal women were analyzed separately. Moreover, to account for a residual effect of age, the two groups were also stratified at 40 years and 60 years, respectively, and the cut-off for HE4 was calculated separately. In classifying malignant gynecological disease vs. non-malignant gynecological diseases and healthy women, dichotomous HE4 showed significantly better accuracy than dichotomous Ca125 for post-menopausal women in both age groups (AUC 0.81 vs. 0.74, *p* = 0.001 for age ≤ 60; AUC 0.78 vs. 0.72, *p* = 0.024 for age > 60) (Figure 1C,D). In contrast, for pre-menopausal women, HE4 did not seem to outperform Ca125 in any age group (Figure 1A,B). These results held true in the classification of malignant gynecological diseases vs. non-malignant gynecological diseases only in post-menopausal state both under and above 60 years (AUC 0.79 vs. 0.73, *p* = 0.006; AUC 0.76 vs. 0.71, *p* = 0.036; respectively) (Figure 2C,D), while for women in premenopausal state over 40 years, dichotomous HE4 showed a higher accuracy than dichotomous Ca125 (AUC 0.73 vs. 0.66, *p* = 0.027) (Figure 2B). The sensitivity analysis showed comparable results in terms of AUC for both Ca125 and HE4 in all scenarios, with the exception of women in post-menopausal state above 60 years, where the HE4 AUC, while still greater, was not significantly different from the Ca125 AUC in classifying malignant gynecological diseases vs. non-malignant gynecological diseases.

## 4. Discussion

The challenge of modern oncology is the integration of different diagnostic data in order to create a clinical picture to better understand all the events related to onset and follow-up of neoplastic disease. Moreover, rapid and accurate diagnosis can be lifesaving for oncologic patients. To this regard, serum biomarkers are more affordable and less invasive than conventional diagnostic imaging examination. To date, OC diagnosis is entrusted to the two biomarkers Ca125 and HE4 [21,22]. Ca125, a mucin, is considered by the FDA to be the gold standard for the management of the OC, although it showed low sensitivity in the early stages of the disease. Increasing levels of this biomarker have been also found in other physiological or pathological conditions, such as menstruation, pregnancy, endometriosis and inflammatory diseases of the peritoneum [23].

Serum human epididymal secretory protein 4 (HE4) has gained relevance as an ovarian cancer biomarker. Because of this, different methods are available for detecting circulating levels of HE4, although several studies have reported the existence of a significant inter-method variability. Thus, harmonization between the different devices should be a pillar for the appropriate clinical use of tumor markers, especially in the case of HE4, since the different threshold of its cut-off in discriminating between benign and malignant ovarian masses is, to date, a very controversial issue [24].

Several diagnostic algorithms have been developed based on the use of both biomarkers [25]. However, the studies carried out have highlighted discrepancy between methods used for the detection of HE4 which, added to the limits of Ca125, often make these algorithms unreliable. Hence, it is important to harmonize the values of the different methods for detecting HE4.

We performed the present study to improve HE4 clinical use and to obtain a reliable evaluation of method agreement for determining whether the use of a common cut-off for different assays may be a viable option.

To this purpose, we compared four modular systems (Abbott Alinity I, Fujirebio Lumipulse G1200 and G600, Roche Cobas 601 and Abbott Architett), which are based on three different immunometric protocols (CMIA, CLEIA and ECLIA).

As a first approach, a common calibration curve was tested by all six participating laboratories, highlighting the presence of analytical bias in the different devices.

In comparison to the reference Lab1, fixed biases were observed in Lab2–4, while a proportional bias was demonstrated in Lab2, 3 and 5. We believe that the systematic biases observed could be ascribed to several interfering factors, such as the different methodology, selectivity of the antibodies and signal detection.

Previous investigation described five variants of HE4 derived by the alternative splicing of the gene and with different tissue expression [24]. It has been suggested that the difference between HE4 assays using different MAbs, at least in some individual samples, could be related to the difference in the ability to detect HE4 variants. Unfortunately, the ability of MAbs to recognize HE4 variants is not well documented, and we can only speculate on the basis of related literature data and conflicting information obtained by manufacturers [17].

Biases identification and the following alignment of the data allowed us to statistically evaluate our database.

Results observed from the comparison of malignant vs. non-malignant and healthy women showed that, in pre-menopausal status, both Ca125 and HE4 are not accurate for the differential diagnosis. Conversely, in the post-menopausal status, HE4 showed a significantly better accuracy than Ca125 for women in both age groups (under and above 60 years) in discriminating malignant from non-malignant pelvic masses. HE4 accuracy was also confirmed comparing malignant gynecological diseases vs. non-malignant gynecological diseases; in this case, in post-menopausal state both under and above 60 years, HE4 showed a statistically significant accuracy with respect to Ca125. Furthermore, as plotted by the ROC curves, HE4 showed a significant accuracy also in pre-menopausal age >40 years, for the differential diagnosis of this malignancy [26]. This evidence could have an helpful implication in the clinical management of women in childbearing age to avoid invasive surgery and preserve fertility, considering that the average age of conceptions has been delayed to 35–40 years [27,28].

According to previous studies, our results confirm the prominent role of HE4 in diagnosis of OC in post-menopausal women [29]. Recently, COVID-19 caused delays in diagnostic investigation, surgical procedures, and routine surveillance for all women. This behavior will have long-lasting consequences, such as later-stage diagnosis and poorer clinical outcomes. In this scenario, it becomes evident that the chance to have a reliable biomarker for the differential diagnosis of ovarian masses is indeed very important. The use of the “biomarker” approach for the assessment of the OC risk can lower the number of patients’ access to hospital facilities, without losing an adequate stratification of malignancy risk to schedule a personalized follow-up. In this context, HE4 could play a key role to better manage subjects at risk of cancer during COVID-19 pandemics.

## 5. Conclusions

As we have extensively discussed, variability in detecting HE4 through different systems is therefore an open and controversial issue. We therefore believe that our analytical approach to correcting detectable biases provides a useful and reliable tool for measuring HE4, and contributes to simplifying the alignment procedures normally recommended for the different instruments used in the laboratories. Thus, in conclusion, in clinical practice we do not suggest dosing HE4 and CA125 in all patients and women indiscriminately, but only in suspected adnexal masses.

## Figures and Tables

**Figure 1 jcm-11-01994-f001:**
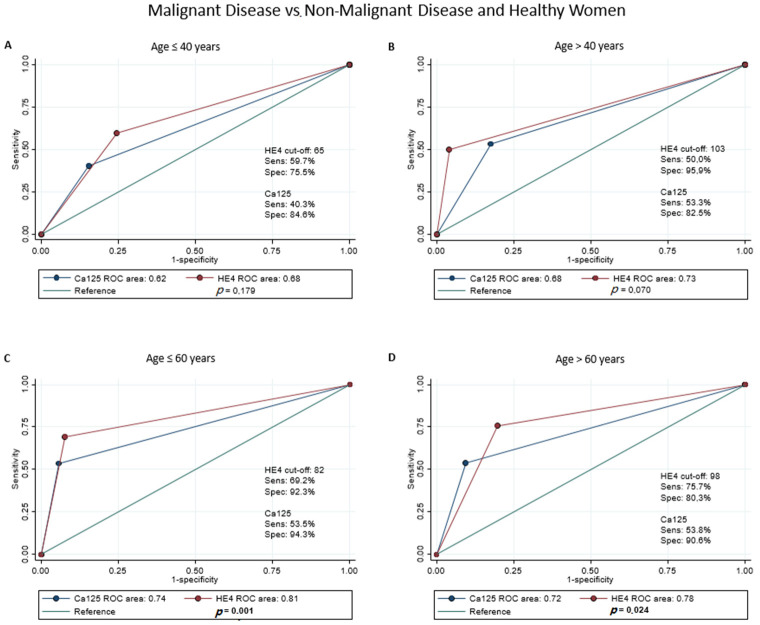
ROC regression analysis of HE4 and Ca125 values in malignant disease vs. non-malignant disease and healthy Women. Population has been stratified as follows: Premenopausal status: ≤40 years (**A**), >40 years (**B**); postmenopausal status: ≤60 years (**C**) and >60 years (**D**).

**Figure 2 jcm-11-01994-f002:**
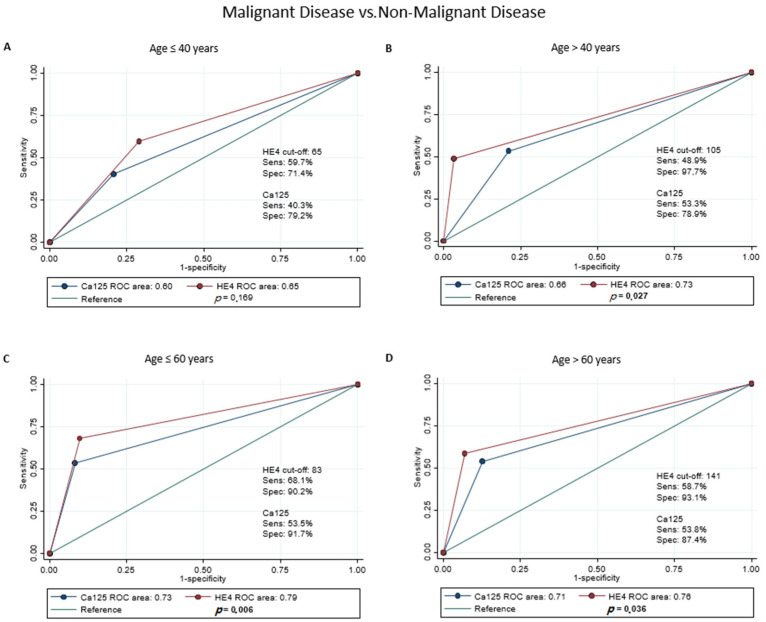
ROC regression analysis of HE4 and Ca125 values in malignant disease vs. non-malignant disease. Population has been stratified as follows: Premenopausal status: ≤40 years (**A**), >40 years (**B**); postmenopausal status: ≤60 years (**C**) and >60 years (**D**).

**Table 1 jcm-11-01994-t001:** Biochemical and physiological properties of the enrolled population. Results are expressed as a number (percentage) or mean (standard deviation); 1232 represents the number of smokers in this female population.

	Malignant Disease Patients *N* = 627 (41.6)	Non-Malignant Disease Patients *N* = 583 (38.6)	Healthy Women *N* = 299 (19.8)
**Age (years), *n* (%)**	59.4 (13.8)	47.3 (13.9)	46.6 (13.0)
**Post-Menopausal Status, *n* (%)**	473 (75.4)	220 (37.7)	144 (48.2)
**Smokers (*n* = 1232), *n* (%)**	124 (26.6)	140 (28.6)	47 (17.1)
**Non-Caucasian Ethnicity, *n* (%)**	12 (1.9)	19 (3.3)	5 (1.7)
**HE4 (pmol/L), mean (SD)**	474.1 (1207.0)	59.7 (62.1)	45.7 (14.7)
**Adjusted HE4 (pmol/L), mean (SD)**	499.62 (1245.6)	68.0 (61.7)	56.3 (14.6)
**CA125 (IU/mL), mean (SD)**	610.4 (7.9)	30.1 (65.9)	15.4 (11.9)
**Creatinine, mean (SD)**	0.8 (0.2)	0.7 (0.2)	0.7 (0.1)

*n*: absolute frequency; %: percentage; SD: standard deviation; pmol: picomoles; IU: international unit.

**Table 2 jcm-11-01994-t002:** Measurement of the calibration curve common to all diagnostic instrumentation. Fixed and proportional biases are expressed as alpha and beta coefficients and their 95% CI. Sample measurements are expressed in pmol/L.

	Lab1	Lab2	Lab3	Lab4	Lab5	Lab6
**Sample A**	1074.4	981.4	927.1	1008.6	1049.0	1027.7
**Sample B**	574.9	497.3	492.5	513.2	501.5	532.1
**Sample C**	384.1	326.4	282.4	347.8	314.4	349.7
**Sample D**	322.4	266.4	244.5	293.8	248.5	239.7
**Sample E**	219.6	183.3	163.4	209.6	174.4	202.3
**Sample F**	186.5	155.9	148.3	175.7	143	169.4
**Sample G**	121.9	105.2	88.8	115.0	95.6	112.7
**Sample H**	88.9	74.55	64.2	82.7	68.7	82.9
**Sample I**	68.0	56.55	51.7	65.4	53.2	60.5
**Sample J**	30.3	25.8	26.4	26.4	28.1	26.8
**Fixed Bias,** **α (95% CI)**	Ref.	**13.9** **(2.3–25.6)**	**21.2** **(3.4–39.0)**	1.88 (−7.64–11.4)	**33.3** **(9.05–57.6)**	13.88 (−7.42–34.96)
**Proportional Bias,** **β (95% CI)**	Ref.	**1.09** **(1.07–1.13)**	**1.15** **(1.10–1.20)**	**1.08** **(1.05–1.10)**	1.02 (0.96–1.08)	1.04 (0.99–1.10)

α: alpha coefficient; β: beta coefficient; CI: confidence interval; Ref.: reference.

**Table 3 jcm-11-01994-t003:** Parametric analysis of the ROC curve regression model with bootstrap sampling.

	Control Model (Linear Model, *n* = 882, Dependent Variable: Adjusted HE4)		ROC Model (Generalized Linear Model with Probit Link, *n* = 1509, 50 Bootstrap Replications)
	β (95% CI)	*p*-Value	β (95% CI)
Creatinine	112.05 (−28.95; 253.05)	0.097	−1.02 (−1.74; −0.29)
Menopausal State (yes)	6.91 (−2.67; 16.49)	0.123	0.52 (0.31; 0.73)

β: Beta Coefficent; CI: Confidence Interval. ROC: Receiving Operator Characteristic.

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
