# Peer review of "New Analytical Approach for the Alignment of Different HE4 Automated Immunometric Systems: An Italian Multicentric Study"

_jcm, 2022, doi:10.3390/jcm11071994_

Round 1

Reviewer 1 Report

The authors have conducted this study with a scientifically rigorous approach.

However, the following suggestions might improve the quality of the manuscript:

  1. The introduction need to be elaborated with clear conceptual frame work and supporting evidences.
  2. Authors have to justify the sample size mterics and how did they arrive at the said sample.
  3. Inlusion and Exclusion criteria have to be clearly mentioned in the manuscript in a detailed manner.
  4. How did they select the serum sample? How did they confirm that these are the samples drawn from the confirmed cancer patients?
  5. Where is the baseline data to confirm the patients with different cancers?
  6. How did you choose the healthy women? 
  7. How have you defined the "healthy women"? what are the base line parameters analysed and to confirm that they are "healthy"?
  8. The discussion is incomplete. Authors are instructed to reqrite the discussion by taking all the essence of the results obtained and supporting with the evidences inorder to make it meaningful.
  9. Statistical analyses part to be reqritten
  10. For example: the statement "Only patients with creatinine values between 0.3 and 3 mg/dl were used in the 204 following analyses." What are those analyses?
  11. Extensive english language editing is warranted for this manuscript.
  12. Conclusion has to be rewritten inorder to convey the study outcome appropriately to the readres.

Author Response

The authors have conducted this study with a scientifically rigorous approach.

However, the following suggestions might improve the quality of the manuscript:

  1. The introduction need to be elaborated with clear conceptual frame work and supporting evidences.

Thank you for this suggestion, introduction have been improved. Changes are highlighted in yellow.

Lines (67-70) To this regard, it has been demonstrated that accurate and early identification of a malignant pelvic mass is essential for an appropriate triage, referral, and subsequent care for women diagnosed with an ovarian malignancy and to improve patient outcomes [4-6].

Lines (85-96) In the last decade, multiple algorithms have been developed to help clinicians in the differential diagnosis between benign or malignant pelvic mass. Among them the Risk of Ovarian Malignancy Algorithm (ROMA) is a logistic regression algorithm that uses HE4 and Ca125 levels along with the patient’s menopausal status to categorize patients into high- and low-risk probabilities that a malignancy will be found in a patient with a pelvic mass . Despite the high sensitivity (94%) and the high negative predictive value (NPV) of 99%, this algorithm showed a specificity of 75% for predicting the presence of epithelial ovarian cancer in women presenting with a pelvic mass [11-13]. In this contest, HE4 alone have been considered more reliable than the ROMA algorithm [7,14]. These observations point out an important issue: the use of HE4, as a circulating biomarker for the diagnosis of OC, alone or in combination with other biomarkers, must be reliable and standardized.

2 Authors have to justify the sample size mterics and how did they arrive at the said sample.

We thank the reviewer for the comment. Sample size section have been added, reporting the metrics used its calculation as follow

Lines (145-149) 2.3 Sample size calculation

The calculation of the sample size was performed assuming an prevalence of malignancy among women with ovarian tumors of 20% [19] and an expected sensitivity of the test of at least 85%, accepting a 95% confidence interval with a width not exceeding 0.05. The resulting sample size was of 980 patients.

3 Inlusion and Exclusion criteria have to be clearly mentioned in the manuscript in a detailed manner.

Thank you for the suggestion, according with the referee we added both criteria.

Lines (155-164) Inclusion criteria: age between 18 and 70 years; for all patients in this study a pelvic mass was confirmed by imaging (i.e. ultrasound, computed tomography, or magnetic resonance) documenting a pelvic mass or ovarian cyst thirty days prior to surgery. All patients underwent surgery for a subsequent tumor histological evaluation and full surgical staging for women diagnoses with a malignancy.  Exclusion criteria: pregnant patients and women under the age of 18 were excluded from the trial. Patients with a prior history of a bilateral salpingoophorectomy were ineligible for the trial. Patients with creatinine levels <0.3 mg/dL and >3mg/dL, with ambiguous gynecological diagnosis, without reported age and classified as in premenopausal stage with age >55 were excluded.

4 -How did they select the serum sample?

All malignancy were confirmed by histological evaluation as reported above in question 3.

-How did they confirm that these are the samples drawn from the confirmed cancer patients?

Age, menopausal status, race, and 67 biomarkers from serum, urine, and plasma samples were preoperatively collected for each patient.

Where is the baseline data to confirm the patients with different cancers?

In agreement with current literature, 90% of Ovarian Cancer are Epithelial Ovarian Cancer.

  1. How did you choose the healthy women?

Ok, we thank the reviewer for the comment. It has been described in methods section

Lines (169-174)  Blood donors without gynecological diseases have been recruited from Trasfusional Unit of the Policlinico Umberto I of Rome

  1. How have you defined the "healthy women"? what are the base line parameters analysed and to confirm that they are "healthy"?

Ok, we thank the reviewer for the comment. It has been described in Methods section

Lines (171-174). “Healthy women were recruited according to the criteria and physical requirements for the selection of blood and blood-components donors following “the Disposition related to the quality requirements and safety of blood and its components”, published in Gazzetta Ufficiale no.69 (2-November 1015).

  1. The discussion is incomplete. Authors are instructed to reqrite the discussion by taking all the essence of the results obtained and supporting with the evidence in order to make it meaningful.

According with your suggestion the discussion has been changed and improved.

  1. Statistical analyses part to be reqritten

We thank the reviewer for the comment. According to the reviewer, we have change it.

  1. For example: the statement "Only patients with creatinine values between 0.3 and 3 mg/dl were used in the 204 following analyses." What are those analyses?

We thank the reviewer for the comment. To clarify the selection of patients involved in the statistical analysis, the exclusion criteria have been removed by the statistical analysis section and properly reported in the database population section:

Lines (155-164): “All subjects included in the data base were between 18 and 70 years; patients with creatine levels <0.3 mg/dL and > 3mg/dL, with ambiguous gynecological diagnosis, without reported age and classified as in premenopausal stage with age >55 were excluded.”

Overall, the statistical analysis section has been improved as follow

Lines (265- 291): “Descriptive statistics were reported using mean and standard deviation for continuous variables and using absolute and relative frequencies for dichotomous and categorical variables. To assess for fixed and proportional systematic biases in HE4 measurement between the various laboratories and Lab1 (reference), the calibration curve results measurements were used to build ordinary least product linear regression models [19]. Briefly, the alpha coefficients whose CI does not contain 0 indicates the presence of a significant fixed bias, while the beta coefficients whose CI does not contain 1 indicates the presence of a significant proportional bias. Measurement from laboratories with significant biases were than adjusted using the appropriate regression equation to correct for such biases.

Factors influencing the accuracy of HE4 in classifying oncology disease were investigate through a parametric analysis of receiver operating characteristic (ROC) curve, using a two steps ROC regression model with bootstrap sampling [20]. Variables initially included in the model were age (continuous) menopausal state (dichotomous) and creatinine levels (continuous) for both the control model and for the ROC estimation model, while the hospitals of origin (categorical) were used as a cluster variable for the bootstrap sampling. Due to high correlation between age and menopausal state, only the latter was kept in the model.

Cut-off level for dichotomization of HE4 was obtained maximizing the Youden index (sum of sensitivity and specificity) against histological classification (Malignant gynecological diseases vs non-malignant gynecological diseases and healthy women or vs non-malignant gynecological diseases alone), rounding the result. For Ca125 dichotomization a standard cut-off of >35 UI/ml was used. ROC curves of dichotomous HE4 and dichotomous Ca125 against histological classification were calculated and the respective area under the curve (AUC) confronted using the test for equality. All analyses were performed using STATA 17.0 (StataCorp LLC, 4905 Lakeway Drive, College Station, 322 Texas, USA). A two-sided p-value <0.05 was considered statistically significant.”

  • Extensive english language editing is warranted for this manuscript.

Thank you for this valuable suggestion. English language was edited accordingly.

  • Conclusion has to be rewritten inorder to convey the study outcome appropriately to the readres.

We thank the reviewer for the comment. According to the reviewer, we have change it.

As we have extensively discussed, variability in detecting HE4 through different systems is therefore an open and controversial issue. We therefore believe that our analytical approach to correcting detectable biases, provides a useful and reliable tool for measuring HE4 and contributes to simplify the alignment procedures normally recommended for the different instruments used in the laboratories.

Reviewer 2 Report

Important topic and valuable impact in the field of oncologic diagnostics. The utility of prognostic and predictive markers is under debate as well as many ongoing studies and analyses are verifying their value. In presented manuscript the Authors described important problem and try to harmonize results/observations. The aim was clearly explained as well methodology is well presented. I have serious doubt concerning study design. 

We perform analyses of different markers because of diagnostic not screening purposes. It means literally in particular case that we suggest blood test to increase our diagnostic accuracy in cases of suspected ovarian masses not in healthy specimens... I.e. The Authors should subdivide enrolled population into 3 groups - 1) ultrasound image of unchanged / healthy ovaries; 2) benign lesions/masses in the ovaries; 3) malignant ovarian tumors (cancers)... Only in such manner we may evaluate value of this marker in perspective of ovarian diagnostics... 

Moreover, the readers may be slightly confused about substantial matter of the study in perspective of the manuscript title vs. methodology/discussion -> you tested ROMA/HE4 values/scores; what about Ca125 (2.6)? 

Funding info should be placed in the proper spot. 

References in perspective of available databases and manuscripts are at least disappointing and surely should be refreshed.

Ref. 8 and 11 are the same. The are missing data, i.e. doi, PMID etc.

There is no need to present map of the Italy in the scientific paper; affiliations are fairly enough.

Missing data/descriptions in the tables; what is what -> %, M+/-SD…

In figures correct <40 into  =<40 i.a.

Editorial mistakes vs or vs. or vs. ; 80 line -> “armonization” etc.

In regard to above-mentioned flaws and concerns as well as having in mind clinical utility of presented topic I recommend essential revision prior subsequent re-assessment.

Author Response

Rew 2

Important topic and valuable impact in the field of oncologic diagnostics. The utility of prognostic and predictive markers is under debate as well as many ongoing studies and analyses are verifying their value. In presented manuscript the Authors described important problem and try to harmonize results/observations. The aim was clearly explained as well methodology is well presented. I have serious doubt concerning study design. 

We perform analyses of different markers because of diagnostic not screening purposes. It means literally in particular case that we suggest blood test to increase our diagnostic accuracy in cases of suspected ovarian masses not in healthy specimens... I.e. The Authors should subdivide enrolled population into 3 groups - 1) ultrasound image of unchanged / healthy ovaries; 2) benign lesions/masses in the ovaries; 3) malignant ovarian tumors (cancers)... Only in such manner we may evaluate value of this marker in perspective of ovarian diagnostics... 

Thank you for this precious suggestion, we included in the same group women with healthy ovaries and with benign lesions and analyzed this group against ovarian cancer since one of the aims of our study is to assess HE4 ability to identify malignancy.

Moreover, the readers may be slightly confused about substantial matter of the study in perspective of the manuscript title vs. methodology/discussion -> you tested ROMA/HE4 values/scores;

Thank for this comment we clarify in the introduction the limits of the ROMA algorithm and the importance of harmonize the intra-methods biases.

what about Ca125 (2.6)? 

 Regarding Ca125 see reference n.8

Funding info should be placed in the proper spot. 

The study was not supported by founding as declared in line 427.

References in perspective of available databases and manuscripts are at least disappointing and surely should be refreshed.

Ref. 8 and 11 are the same. The are missing data, i.e. doi, PMID etc.

We thank the reviewer for this precious comment. We refreshed the references accordingly.

There is no need to present map of the Italy in the scientific paper; affiliations are fairly enough.

We thank the reviewer for this suggestion. We eliminated the map accordingly.

Missing data/descriptions in the tables; what is what -> %, M+/-SD?

We apologize, but we couldn’t find missing information in the table. Could you please indicate the correct line/s or table?

We thank the reviewer for this comment. We added abbreviations below the tables.

In figures correct <40 into  =<40 i.a.

Thank you for this precious suggestion. We corrected the figures accordingly.

Editorial mistakes vs or vs. or vs. ; 80 line -> ?armonization? etc.

Thank for this comment, we changed the typing errors.

In regard to above-mentioned flaws and concerns as well as having in mind clinical utility of presented topic I recommend essential revision prior subsequent re-assessment.

Round 2

Reviewer 1 Report

Authors have addressed all the queries of the reviewers.

Author Response

We perform analyses of different markers because of diagnostic not screening purposes. It means literally in particular case that we suggest blood test to increase our diagnostic accuracy in cases of suspected ovarian masses not in healthy specimens... I.e. The Authors should subdivide enrolled population into 3 groups - 1) ultrasound image of unchanged / healthy ovaries; 2) benign lesions/masses in the ovaries; 3) malignant ovarian tumors (cancers)... Only in such manner we may evaluate value of this marker in perspective of ovarian diagnostics...

Thank you for this precious suggestion, we included in the same group women with healthy ovaries and with benign lesions and analyzed this group against ovarian cancer since one of the aims of our study is to assess HE4 ability to identify malignancy.

This is insufficient explanation... We do not suggest performance of HE4/Ca125/ROMA in patients e.g. with endometrial fibroid or myoma because it has do diagnostic value; moreover we do not perform population screening with above mentioned markers so those markers are reserved for ovarian mass testing to increase diagnostic accuracy... If we have patients with healthy ovaries and endometrial fibroid should be classified for healthy specimens not benign group etc. I'm not convinced and in my opinion it should be changed as it is confusing and will have impact on results...

Thank you for bringing our attention to this point , We thank the reviewer for the comment. To address the issue, we performed a sensitivity analysis considering as benignant patients only women with ovarian, annexal and peritoneal lesions, as well as patients with endometriosis, accounting for 411 of the original 583 benignant patients. The analysis showed comparable results in terms of AUC for both Ca125 and HE4. The Methods and the Results section have been modified as follow:

Pag 9 “A sensitivity analysis was performed considering as non-malignant gynecological diseases only ovarian, annexal and peritoneal lesions, and endometriosis.” (highlighted in green ).

Pag 11 “The sensitivity analysis showed comparable results in terms of AUC for both Ca125 and HE4 in all scenarios, with the exeption of women in post-menopausal state above 60 years, were the HE4 AUC, while still greater, was not significantly different from the Ca125 AUC in classifing malignant gynecological diseases versus non-malignant gynecological diseases.” (highlighted in green ).

Pag 15 to better clarify our study we added in the conclusion this sentence (highlighted in green ).

what about Ca125 (2.6)?

Regarding Ca125 see reference n.8

This insufficient explanation... I realize where is reference for particular marker... You include results from different laboratory as you mentioned... In general this is ok for comparison purposes, but there is only short remark that the same machines/equipment for tests was used... In such character of the study it is extremely important... Especially to show what was compared/combined with what... 

Pages 7-8, As suggested, we have revised the CA125 section by introducing the equipment for tests  used (highlighted in green ).

Funding info should be placed in the proper spot.

The study was not supported by founding as declared in line 427.

Pag 15 We thank you for highlighting our mistake , This study is founded by The University of Rome “Sapienza”. highlighted in green ).

Missing data/descriptions in the tables; what is what -> %, M+/-SD?

We apologize, but we couldn’t find missing information in the table. Could you please indicate the correct line/s or table?

Especially in table 1. 

Pag 9 -10 We thank the reviewer for the comment. We added the description of the measurements in (table 1 and table 2.)

Reviewer 2 Report

We perform analyses of different markers because of diagnostic not screening purposes. It means literally in particular case that we suggest blood test to increase our diagnostic accuracy in cases of suspected ovarian masses not in healthy specimens... I.e. The Authors should subdivide enrolled population into 3 groups - 1) ultrasound image of unchanged / healthy ovaries; 2) benign lesions/masses in the ovaries; 3) malignant ovarian tumors (cancers)... Only in such manner we may evaluate value of this marker in perspective of ovarian diagnostics...

Thank you for this precious suggestion, we included in the same group women with healthy ovaries and with benign lesions and analyzed this group against ovarian cancer since one of the aims of our study is to assess HE4 ability to identify malignancy.

This is insufficient explanation... We do not suggest performance of HE4/Ca125/ROMA in patients e.g. with endometrial fibroid or myoma because it has do diagnostic value; moreover we do not perform population screening with above mentioned markers so those markers are reserved for ovarian mass testing to increase diagnostic accuracy... If we have patients with healthy ovaries and endometrial fibroid should be classified for healthy specimens not benign group etc. I'm not convinced and in my opinion it should be changed as it is confusing and will have impact on results...

what about Ca125 (2.6)?

Regarding Ca125 see reference n.8

This insufficient explanation... I realize where is reference for particular marker... You include results from different laboratory as you mentioned... In general this is ok for comparison purposes, but there is only short remark that the same machines/equipment for tests was used... In such character of the study it is extremely important... Especially to show what was compared/combined with what... 

Funding info should be placed in the proper spot.

The study was not supported by founding as declared in line 427.

This study is founded by The University of Rome “Sapienza”.

Missing data/descriptions in the tables; what is what -> %, M+/-SD?

We apologize, but we couldn’t find missing information in the table. Could you please indicate the correct line/s or table?

Especially in table 1. 

Author Response

(The authors gave the same response as above.)
